·ᐧPLOS | ONE

# Assessing the performance of genome-wide association studies for predicting disease risk

**Jonas Patron**[1], **Arnau Serra-Cayuela**[1], **Beomsoo Han**[1], **Carin Li**[1],
**David Scott Wishart**[1,2]*

**1** Department of Biological Sciences, University of Alberta, Edmonton, Canada, **2** Department of Computing Science, University of Alberta, Edmonton, Canada

* dwishart@ualberta.ca

**Data Availability Statement:** The code is freely available for download at https://github.com/jonaspatronjp/GWIZ-Rscript/, and is compatible with a wide variety of UNIX platforms, Mac OS and Windows operating systems. The GitHub download file also includes an example of the *.csv input file,

## Abstract

To date more than 3700 genome-wide association studies (GWAS) have been published that look at the genetic contributions of single nucleotide polymorphisms (SNPs) to human conditions or human phenotypes. Through these studies many highly significant SNPs have been identified for hundreds of diseases or medical conditions. However, the extent to which GWAS-identified SNPs or combinations of SNP biomarkers can predict disease risk is not well known. One of the most commonly used approaches to assess the performance of predictive biomarkers is to determine the area under the receiver-operator characteristic curve (AUROC). We have developed an R package called G-WIZ to generate ROC curves and calculate the AUROC using summary-level GWAS data. We first tested the performance of G-WIZ by using AUROC values derived from patient-level SNP data, as well as literature-reported AUROC values. We found that G-WIZ predicts the AUROC with <3% error. Next, we used the summary level GWAS data from GWAS Central to determine the ROC curves and AUROC values for 569 different GWA studies spanning 219 different conditions. Using these data we found a small number of GWA studies with SNP-derived risk predictors that have very high AUROCs (>0.75). On the other hand, the average GWA study produces a multi-SNP risk predictor with an AUROC of 0.55. Detailed AUROC comparisons indicate that most SNP-derived risk predictions are not as good as clinically based disease risk predictors. All our calculations (ROC curves, AUROCs, explained heritability) are in a publicly accessible database called GWAS-ROCS (http://gwasrocs.ca). The G-WIZ code is freely available for download at https://github.com/jonaspatronjp/GWIZ-Rscript/.

## Introduction

A genome-wide association study (GWAS) is a comprehensive genetic analysis of the association between certain observable traits and specific genetic variations in the form of Single Nucleotide Polymorphisms (SNPs). The appeal of genome wide association (GWA) studies is that they provide a relatively facile approach for detecting potential genetic contributors to common, complex diseases (such as diabetes) or phenotypes (such as body mass index or hair color) using a simple case-control study model. The first GWA study was performed in 2005

as well as instructions on how to run the program. In addition, the complete set of simulated populations used in this study, along with their corresponding SNP profiles, disease status, calculated ROC curves and AUROC data is available for download at http://gwasrocs.ca.

**Funding:** Funding for this research has been provided by Genome Canada [https://www.genomecanada.ca/] (to DSW), Genome Alberta [http://www.genomealberta.ca/] (to DSW), the Canadian Institutes of Health Research [http://www.cihr-irsc.gc.ca/e/193.html] (to DSW), the Canada Foundation for Innovation [https://www.innovation.ca/] (to DSW) and the Natural Sciences and Engineering Research Council [http://www.nserc-crsng.gc.ca/index_eng.asp] (to DSW). The funders had no role in study design, data collection and analysis, decision to publish, or preparation of the manuscript.

**Competing interests:** The authors have declared that no competing interests exist.

[1]. This early work explored the association between certain SNPs and age-related macular degeneration in a study population of 146 individuals (96 cases, 50 controls). To date, thousands of GWA studies looking at almost an equal number of conditions or phenotypes, with study populations as large as 1.3 million have been published [2]. Many of these GWA studies are now archived in public databases such as GWAS Central [3] and the NHGRI-EBI GWAS Catalog [4].

Public databases such as GWAS Central contain summary level findings from GWA studies collected on humans [3]. GWAS Central currently houses data from more than 3300 publications corresponding to over 6100 GWA studies, and lists more than 21 million p-values, ranging between $5 \times 10^{-2}$ and $1 \times 10^{-584}$. The p-values in a GWA study report the likelihood of the odd-ratios between two different alleles being statistically different than one. The typical threshold of significance for most published GWA studies is $p = 5 \times 10^{-8}$. The average odds ratio for a statistically significant SNP is 1.33 with very few SNPs having an odds ratio above 3.0 [5].

GWA studies differ from most other 'omics or clinical/epidemiological studies in their method of reporting of disease-associated or disease risk markers. In particular, relatively strong emphasis is placed on the reporting of p-values of a single SNP marker (with p-values of $< 10^{-50}$ often being achieved) along with its associated odds ratio [6]. Interestingly, very few GWA studies combine multiple SNPs together to produce a multi-marker risk predictor. As a result, it is rare to find a GWA study that reports on the sensitivity, specificity, receiver operating characteristic (ROC) curve or the area under the ROC curve (AUROC or C-statistic) associated with a given SNP or combination of SNPs for disease risk or trait prediction. In contrast to most GWA studies, clinical, proteomic and metabolomic studies involving marker identification rarely achieve p-values of $< 10^{-6}$ and they infrequently report the performance of their markers in terms of odds ratios. Instead, most marker-based clinical, proteomic or metabolomic studies tend to combine multiple clinical or 'omic measures to generate multi-marker risk predictors. In these studies, multi-marker sensitivity, specificity, ROC curves and/or AUROCs are routinely reported [7].

One way for the different 'omics communities to better understand the predictive or discriminative ability of GWAS data is to convert the reported SNP data into a more conventional biomarker reporting format. In particular, if multi-SNP biomarker data could be combined and converted into ROC or AUROC data, then a more direct comparison could be performed regarding the performance of SNPs for predicting disease risks relative to clinical, metabolite or protein biomarkers for the same conditions. The only precise way of generating ROC curves or calculating AUROC values from GWA studies is to have the complete patient SNP data set. Unfortunately, obtaining ethics approvals and permissions for access to the full patient data sets from thousands of individual GWA studies (found in GWAS Central, for example), conducted in multiple countries and spanning more than 15 years is currently impractical.

Over the past decade several approaches to calculating ROC curves and AUROC data from summary level SNP data have appeared [8–13]. Unfortunately, we found that these methods were not particularly accurate, very limited in their capabilities or required more information than what was available in standard GWAS Central summary data (see Results for more details). To overcome these issues we developed a novel approach to accurately generate ROC curves and to calculate the AUROC for different SNP combinations using the summary-level data that is standardly found in GWAS databases. Summary level data, which only contains study-wide averages, p-values, odds ratios, risk allele frequencies and other summary statistics for the entire study population, is the only GWAS data can be deposited in public repositories such as GWAS Central. Our new method (called G-WIZ) combines population modeling with logistic regression (for risk prediction) to generate study-specific ROC curves and AUROC

values. Our approach also enables the estimation of SNP heritability directly from summary level GWAS data. We have extensively tested G-WIZ by measuring its ability to predict ROC curves and AUROC values derived from a large collection of authentic individual patient SNP data (i.e. non-summary data), as well as to predict literature-reported AUROC values. We found that G-WIZ predicts the AUROC with <3% error, which is much better than any other method published to date. We then used G-WIZ to, calculate ROC curves and AUROC values for 569 GWA studies spanning 219 different conditions/phenotypes using summary level GWAS data from GWAS Central. Using these data we found at least five conditions/phenotypes that exhibit very high AUROCs (>0.75) and SNP-heritability values (>30%) using a multi-component SNP risk predictor. On the other hand, we found that the average AUROC value for GWAS risk predictors is only 0.55. In contrast, most predictive clinical, metabolite or protein-based risk predictors have AUROCs of >0.7 or 0.8 [7,14]. We also used the data from these calculations to derive a simple formula to estimate SNP heritability directly from the AUROC value. All of the G-WIZ calculations and accompanying analyses (ROC curves, AUROCs, explained heritability, etc.) have been deposited into a publicly accessible database called GWAS-ROCS (http://gwasrocs.ca). Having such a large, centralized database of SNP combinations, ROC curves, AUROC values and heritability estimates should open the door to performing more systematic comparisons of GWA studies or to identify new and unexpected trends or novel disease-gene relationships.

## Methods

### Ethics approvals

We received research ethics approval from the University of Alberta (REB—Pro00084706) and approval from the Wellcome Sanger Institute (request ID: 8303 and 7104) to obtain access to the Wellcome Trust Case Control Consortium (WTCCC) GWAS datasets [15].

### Selection of GWAS Central studies

An in-house Python script was created to screen-scrape the summary data in GWAS Central. A total of 3307 GWAS publications corresponding to 6137 GWAS summaries were collected in this manner. These GWA studies were further filtered based on the inclusion of an odds ratio (OR) and a risk allele frequency (RAF) for each reported SNP. Additionally, to improve AUROC value prediction consistency we discarded GWA studies with a sample size of less than 1000 cases or 1000 controls. We thought this was reasonable since the WTCCC datasets all had sample sizes of more than 1000 cases and more than 1000 controls. Moreover, an analysis of every study in GWAS Central indicated that roughly 70% of all published GWA studies (since 2009) have had sample sizes of greater than 1000 cases and 1000 controls. While choosing this threshold should have left us with about 4300 studies to analyze, we found that many studies in GWAS Central were missing either the SNP odds ratios or the risk allele frequencies–or both, which prevented their use in our calculations. Indeed, after applying these filters, we were left with a total of 569 GWAS Central studies, corresponding to 219 different conditions.

### Population simulation for ROC and AUROC calculations

The calculation of sensitivity, specificity, ROC and AUROC data from case/control studies normally requires a data set where the variables of interest (SNPs, proteins, metabolites, clinical measures, etc.) are assigned to individual patients along with their health status. Unfortunately, the GWAS data as found in GWAS Central or other public databases, is only available

as summary data. This means that this patient-specific information is not readily accessible. Indeed, the only information retrievable is the cohort size (cases and controls), the SNP identifiers, the p-values, ORs for each SNP and RAFs for each reported SNP. While it is possible to obtain detailed SNP profiles and health status information for each patient in each study, doing so for the entire GWAS Central Collection would have required extensive ethics reviews along with time and resources that were far beyond our means. This necessitated the development of a modeling program (called G-WIZ or Gwas WIZard) that would calculate ROC and AUROC data from summary-level information only. To do so we exploited that fact that almost no SNPs separated by more than 10 kb are in absolute linkage disequilibrium and that the vast majority of reported, disease-significant SNPs are in Hardy-Weinberg equilibrium [16–18]. As a result, we assumed the independence of SNPs to create simulated patient populations with specific SNP profiles and assigned health conditions from the publicly available OR and RAF data. These synthetic populations were designed to be sufficiently large (typically >30,000 individuals) so that statistically anomalies would be averaged out. To assign a single SNP to an individual in the simulated population the following methodology was used:

Let $H_s$ denote the number of risk alleles in the healthy group,

$H_n$ denote the number of non-risk alleles in healthy group,

$D_s$ denote the number of risk alleles in the diseased group,

$D_n$ denote the number of non-risk alleles in diseased group,

*RAF* denote risk allele frequency,

*OR* denote odds ratio,

$N_{control}$ denote the number of controls in the simulated dataset,

$N_{case}$ denote the number of cases in the simulated dataset,

$\lceil \cdot \rceil$ denote the ceiling function.

Using $N_{control}$, $N_{case}$, *OR* and *RAF* as the pre-specified input values, we can calculate $H_s$, $H_n$, $D_s$, $D_n$ as follows:

$$H_s = \lceil 2 \cdot N_{control} \cdot RAF \rceil$$

$$H_n = 2 \cdot N_{control} - H_s$$

$$D_s = \left\lceil \frac{2 \cdot OR \cdot N_{case} \cdot H_s}{H_n + OR \cdot H_s} \right\rceil$$

$$D_n = 2 \cdot N_{case} - D_s$$

Under the assumption that each SNP is independent of the others we can repeat the above procedure to create a full SNP profile and an assigned health state for each member of the simulated population. More specifically, a G-WIZ simulation starts by creating a population of individuals assigned as cases or controls in accordance with the selected GWAS Central record. Next, by using the risk allele frequency for the controls and the odds ratio between the cases and controls G-WIZ calculates the risk allele frequency in the cases. Once the risk allele frequency in both the case and control groups is generated, G-WIZ can appropriately assign the SNP profiles to each group. All G-WIZ models were built using all the SNPs reported by each of the respective GWA studies. These SNPs from GWAS Central were reported on the basis of their significance as identified by the original depositors. However, we considered that it might still be possible that models created using only a subset (feature selection) of reported SNPs would perform better, as this would have controlled for over-parameterization. We tested for this by performing feature selection, using only the SNPs with lowest p-values,

however no improvements to the models' performance were found. In the end our SNP profiles used every reported SNP. On average, each study had a SNP profile consisting of 6 significant SNPs. Moreover, the maximum SNP p-value was $9 \times 10^{-6}$ and the minimum SNP p-value was $1 \times 10^{-295}$, indicating that the reported SNPs are all highly significant. Further, each G-WIZ model had on average 34,491 simulated patients (cases and controls). The largest number of SNPs used in any given SNP profile was 50, and the largest synthetic population generated by G-WIZ consisted of 808,380 individuals.

## Statistical modeling for ROC curve generation

The creation of simulated populations consisting of full SNP profiles and assigned health states for each of the 569 condition/phenotype studies in GWAS Central allowed us to calculate the corresponding ROC curves and AUROC values. A common modeling method used to generate ROC curves for multi-marker data is logistic regression. Logistic regression is a statistical method for modeling multiple independent variables (e.g. SNPs) to explain two possible outcomes (e.g. healthy or diseased). Once constructed, a logistic regression model will return a risk score between 0 and 1. A cut-off value can then be chosen (e.g. 0.5), and any individual that has a risk score above it is classified as 'diseased', and any individual below it is classified as 'healthy'. A plot of the sensitivity against 1-specificity for all possible cut-off values is known as a receiver-operating characteristic (ROC) curve. The classification accuracy of the logistic regression model can then be measured by calculating the area under the curve of the ROC curve (AUROC). A perfect model would have an AUROC of 1, while a model with no classification accuracy would have an AUROC of 0.5 [7].

We performed logistic regression analysis because it is easy to perform and interpret [19]. Although, it is possible that better performing multi-SNP profiles could have been developed using advanced machine learning algorithms such as neural networks, decision trees, or support vector machines [20–26], it is also possible to over-train models with these very powerful pattern-finding tools. Indeed, it is not unusual, during validation studies, to see these models fall short when compared to logistic regression models [20–23]. These concerns regarding overfitting led us to limit our model complexity and to exclusively use logistic and ridge logistic regression to estimate the classification accuracy of these GWA studies.

Another common issue with regression models containing many explanatory variables is multicollinearity. Multicollinearity increases the variance of parameter estimates, which will affect confidence intervals and hypothesis tests. This can lead to incorrect inferences about relationships between explanatory and response variables [27]. With these issues in mind we tested for multicollinearity by estimating the variance inflation factor (VIF) prior to building each regression model. The VIF is the quotient of the variance from a model which regresses one of the predictor variables against all the others. Multicollinearity was determined to exist when at least two variables showed an inflated coefficient (i.e. when the VIF was infinity). We tried a wide range of other VIF cutoff values (less than infinity), however the differences in the AUC estimates were very small (<0.009). Whenever multicollinearity was observed we used ridge logistic regression [28] to generate a biomarker model, otherwise we used a standard logistic regression model. Because standard logistic regression is more easily interpretable than its ridge regression counterpart, we found it appropriate to restrict the use of ridge regression only to models with extreme (i.e. divergent) VIF estimates. In total 566 standard logistic regression models and 3 ridge logistic regression models were constructed for all 569 GWA studies.

To assess the performance of each biomarker or ROC-generative model, the simulated data was randomly split into training and testing sets. In the training set, nested cross-validation

(outer 3-fold and inner 2-fold) was used to obtain an estimate of the classification accuracy [28]. Once the model was properly tuned, it was validated using the testing set.

## The G-WIZ program

G-WIZ is written in the R programming language [29]. It consists of several modules including a custom-written tool to generate patient populations, the MLR package [30] to build the logistic regression models and perform cross-validation, as well as the pROC package [31] to generate ROC curves and to calculate the AUROCs. To analyze a study with G-WIZ the user must create a $^*$.csv file with the study sample size, and the odds ratios and risk allele frequencies for each SNP in the study. A typical study (with a sample size of 50,000 individuals) can be simulated and analyzed via G-WIZ on a modern laptop computer in less than 30 seconds. For any given study, G-WIZ performs population simulation, regression modelling, and then estimates ROC curves and AUROCs. The output consists of a $^*$.csv file with the regression coefficients, a $^*$.csv file with the sensitivity, specificity and AUROC of the study, a $^*$.csv file with the simulated population, and a $^*$.png file with the ROC curve. The code is freely available for download at https://github.com/jonaspatronjp/GWIZ-Rscript/, and is compatible with a wide variety of UNIX platforms, Mac OS and Windows operating systems. The GitHub download file also includes an example of the $^*$.csv input file, as well as instructions on how to run the program. In addition, the complete set of simulated populations used in this study, along with their corresponding SNP profiles, disease status, calculated ROC curves and AUROC data is available for download at http://gwasrocs.ca.

## SNP-derived heritability calculations

For each study in GWAS Central we also estimated the total variance in disease liability explained (often referred to as the SNP heritability) using the following formula described by Pawitan et al. [32].

Let *var(·)* denote the variance
*OR* denote the odds ratio
*log(·)* denote the natural logarithm
*RAF* denote the risk allele frequency
$\pi$ denote the mathematical constant pi
$h^2$ denote the heritability
*g* denote the random genetic effect
Then

$$h^2 = \frac{var(g)}{\text{var}(g) + \pi^2/3}$$

$$var(g) = 2\sum_{k=1}^{n} RAF_k(1 - RAF_k)(logOR_k)^2$$

Where *n* is the number of SNPs in a particular study, and $RAF_k$ and $OR_k$ are the risk allele frequency and odds ratio of the $k^{th}$ SNP. We created an in-house R script to run this formula on all 569 GWA studies collected from GWAS Central. The results are shown in S1 Table.

## Validation

To further ensure that our modeling methods and assumptions were correct, we validated our predictions in two different ways. In the first approach, we used real patient GWAS data from the Wellcome Trust Case Control Consortium (WTCCC). Using this data we ran logistic

regression analyses for 7 different conditions on 2 different control datasets, and calculated the true ROCs and AUROCs. The 7 conditions were bipolar disorder (BD), coronary artery disease (CAD), Crohn's disease (CD), hypertension (HT), rheumatoid arthritis (RA), type 1 diabetes (T1D), and type 2 diabetes (T2D)—spanning AUROC values from 0.51 to 0.72. The two control datasets were WTCCC1 project samples from UK National Blood Service and WTCCC1 project samples from 1958 British Birth Cohort. The SNP profiles and risk alleles we used are reported in Table 1, and are the same SNPs and alleles reported as being statistically significant by the WTCCC researchers [15]. For both control datasets we then applied the G-WIZ modelling method to the same set of SNPs and generated a synthetic population with SNP profiles by directly calculating the RAFs and number of cases and controls for each disease study, and by estimating the ORs from the logistic regression coefficients. The true AUROCs were then compared to our G-WIZ calculated AUROCs. Additionally, the shape of the ROC curves was also compared using Delong's test [33].

The second approach we used to validate our G-WIZ ROC and AUROC predictions involved comparing the G-WIZ predictions against 21 AUROC values provided by previously published GWA studies [34–54]. The GWA studies selected for this validation test covered a broad range of conditions/phenotypes, and a wide range of AUROC values (0.55–0.74). The same set of SNPs reported for these studies were used in the G-WIZ summary-level

**Table 1. SNP profiles and risk alleles reported by the WTCCC publishers.**

| WTCCC Disease | SNP | Risk Allele | Minor allele |
|---|---|---|---|
| Bipolar disorder | rs420259 | A | G |
| Coronary artery disease | rs1333049 | C | C |
| Crohn's disease | rs11805303 | T | T |
| Crohn's disease | rs10210302 | T | C |
| Crohn's disease | rs9858542 | A | A |
| Crohn's disease | rs17234657 | G | G |
| Crohn's disease | rs1000113 | T | T |
| Crohn's disease | rs10761659 | G | A |
| Crohn's disease | rs10883365 | G | G |
| Crohn's disease | rs17221417 | G | G |
| Crohn's disease | rs2542151 | G | G |
| Hypertension | rs2820037 | T | T |
| Hypertension | rs6997709 | G | T |
| Hypertension | rs7961152 | A | A |
| Hypertension | rs11110912 | G | G |
| Hypertension | rs1937506 | G | A |
| Hypertension | rs2398162 | A | G |
| Rheumatoid arthritis | rs6679677 | A | A |
| Rheumatoid arthritis | rs6457617 | T | T |
| Type 1 diabetes | rs6679677 | A | A |
| Type 1 diabetes | rs9272346 | A | G |
| Type 1 diabetes | rs11171739 | C | C |
| Type 1 diabetes | rs17696736 | G | G |
| Type 1 diabetes | rs12708716 | A | G |
| Type 2 diabetes | rs9465871 | C | C |
| Type 2 diabetes | rs4506565 | T | T |
| Type 2 diabetes | rs9939609 | A | A |

simulations. We used the published ORs, RAFs and the number of cases and controls reported in the 21 papers. Logistic regression or ridge regression (as required) was used to generate the ROC curves and to calculate the AUROCs for each of the 21 studies. The reported AUROCs were then compared to our calculated AUROCs. To compile these 21 studies, we created a list of all applicable papers in PubMed by searching the literature for all publications with genomic prediction models that reported an AUROC. This gave us a list of 112 studies (see S2 Table). Of these 112 studies, we further filtered the studies by study design, sample size requirements, and available SNP information. This left us with 21 studies that were both case-control studies, that had more than 1000 cases and 1000 controls, and that had the relevant SNP summary data to run G-WIZ.

## Results

### Validation testing

A crucial part of developing G-WIZ was to perform an extensive set of validation tests to ensure that our AUROC predictions were accurate. We tested the performance of G-WIZ with respect to four different SNP inheritance models (dominant, recessive, odd SNPs are dominant, and odd SNPs are recessive). This was important because the SNP inheritance model in published GWA studies may vary from SNP to SNP, and is not typically disclosed. Thus, for each of the 7 WTCCC disease studies we calculated the true AUROC under these four SNP inheritance schemes. The results are outlined in Table 2. In total, we ran 56 different tests (spanning all 7 different disease studies, 2 control datasets and 4 SNP inheritance schemes). The average difference in our AUROC predictions was 0.026. The average p-value from the Delong's test was 0.43, indicating that the predicted ROC curves did not differ significantly from the true ROC curves. Indeed, 52 out of 56 tests had ROC curves that did not differ significantly when tested at the $p = 0.05$ level of significance.

**Table 2. Comparison of true versus G-WIZ predicted AUROCs for the WTCCC data.**

| Cases Cohort | Controls Cohort: WTCCC1 UK National Blood Service | | |
|---|---|---|---|
| | True AUROC* | Predicted AUROC* | Difference |
| WTCCC1 Bipolar disorder | 0.52 | 0.52 | 0.00 |
| WTCCC1 Coronary artery disease | 0.53 | 0.56 | 0.03 |
| WTCCC1 Chron's disease | 0.63 | 0.62 | 0.01 |
| WTCCC1 Hypertension | 0.57 | 0.57 | 0.00 |
| WTCCC1 Rheumatoid arthritis | 0.61 | 0.62 | 0.01 |
| WTCCC1 Type 1 diabetes | 0.68 | 0.70 | 0.02 |
| WTCCC1 Type 2 diabetes | 0.57 | 0.57 | 0.00 |
| Cases Cohort | Controls Cohort: WTCCC1 1958 British Birth Cohort | | |
| | True AUROC* | Predicted AUROC* | Difference |
| WTCCC1 Bipolar disorder | 0.52 | 0.52 | 0.00 |
| WTCCC1 Coronary artery disease | 0.54 | 0.57 | 0.03 |
| WTCCC1 Chron's disease | 0.64 | 0.60 | 0.04 |
| WTCCC1 Hypertension | 0.55 | 0.55 | 0.00 |
| WTCCC1 Rheumatoid arthritis | 0.61 | 0.61 | 0.00 |
| WTCCC1 Type 1 diabetes | 0.66 | 0.71 | 0.05 |
| WTCCC1 Type 2 diabetes | 0.56 | 0.56 | 0.00 |

*Values are the average calculated over the 4 different SNP inheritance models dominant, recessive, dominant (odd)—recessive (even), dominant (even)—recessive (odd)

**Table 3. Comparison of published versus G-WIZ predicted AUROCs for 21 publications.**

| Phenotype/Condition | Published AUROC [Reference) | Predicted AUROC | Difference |
|---|---|---|---|
| Type 2 diabetes | 0.58 [36] | 0.59 | 0.01 |
| Type 2 diabetes | 0.60 [34] | 0.62 | 0.02 |
| Type 2 diabetes | 0.63 [37] | 0.62 | 0.01 |
| Type 2 diabetes | 0.60 [35] | 0.58 | 0.02 |
| Type 2 diabetes | 0.62 [39] | 0.58 | 0.04 |
| Type 2 diabetes | 0.63 [40] | 0.60 | 0.03 |
| Coronary artery disease | 0.61 [41] | 0.58 | 0.02 |
| Psoriasis | 0.72 [42] | 0.67 | 0.05 |
| Rheumatoid arthritis | 0.59 [43] | 0.58 | 0.01 |
| Breast cancer | 0.58 [44] | 0.63 | 0.05 |
| Colorectal cancer | 0.57 [38] | 0.60 | 0.03 |
| Breast cancer | 0.58 [45] | 0.61 | 0.03 |
| Prostate cancer | 0.66 [46] | 0.65 | 0.01 |
| Lung cancer | 0.55 [47] | 0.52 | 0.04 |
| Esophageal squamous-cell carcinoma | 0.63 [48] | 0.63 | 0.00 |
| Rheumatoid arthritis | 0.66 [49] | 0.69 | 0.03 |
| Venous thrombosis | 0.66 [50] | 0.64 | 0.02 |
| Alzheimer's disease | 0.70 [51] | 0.77 | 0.07 |
| Colorectal cancer | 0.56 [52] | 0.59 | 0.03 |
| Leprosy | 0.74 [53] | 0.80 | 0.06 |
| Glaucoma | 0.62 [54] | 0.57 | 0.05 |

Table 3 lists the AUROC values calculated by our logistic regression modeling technique along with the AUROCs from the 21 previously published GWAS papers. These published AUROC values were derived from GWA studies that used patient-identifiable, authentic clinical data. As can be seen from this table, our AUROC predictions had an average difference of 0.030 relative to the published AUROCs, with a maximum difference of 0.071 and a minimum difference of 0.004. This is remarkably good considering that our data consisted of simulated populations while the published studies were working with complete population data. Previously published studies on AUROC estimates have shown that the type of modeling method used to generate a multi-component biomarker signature can lead to AUROC variations of +/- 0.07 [20–23]. Similarly, we observed that the size of the sample population can also lead to differences in AUROC values (of +/- 0.04) with smaller populations or unbalanced numbers of cases and controls leading to larger differences.

The excellent performance of our G-WIZ AUROC estimates in this validation study (along with the robustness shown in the cross-validation tests) gave us sufficient confidence to apply our logistic modeling method to all 569 GWA studies compiled from GWAS Central and calculate their ROC curves and AUROCs.

## Analysis of GWAS Central studies

The complete set of ROCs, along with the calculated AUROCs for all 569 GWAS studies is available at http://gwasrocs.ca. These datasets were used to calculate the plot shown in Fig 1. This figure displays a histogram of the calculated AUROC distribution for all 569 GWAS Central studies. The average AUROC was 0.55 while the median AUROC was 0.54.

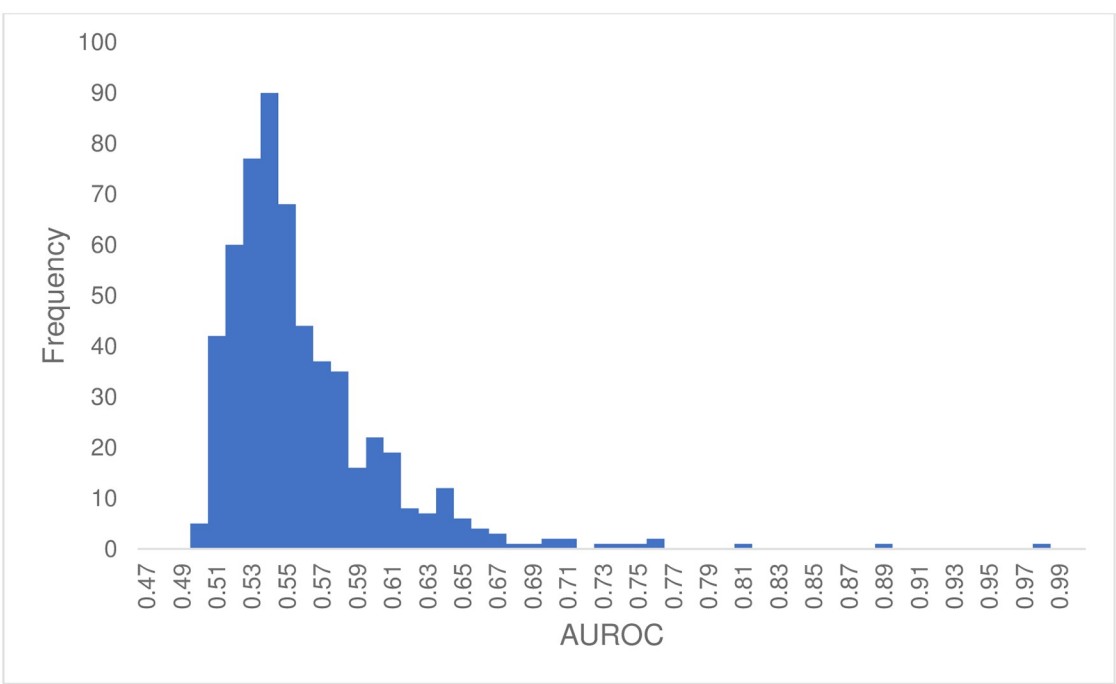

**Fig 1. Histogram of the G-WIZ predicted AUROCs of the 569 studies from GWAS Central.** Bin width of 0.01.

Ten GWA studies (2%) had an AUROC greater than 0.70 and 73 (13%) had an AUROC above 0.60. The GWA study with the highest AUROC was "Black vs. non-black hair color" [55,56], which had a predicted AUROC of 0.98. The GWA study "Shingles" [57,58] had the lowest AUROC with a predicted AUROC of 0.50. The logistic regression models that we built used on average 6 SNPs. The largest number of SNPs used in a single model was 50. Six studies were modeled using 50 SNPs and 165 studies (29%) were modeled using only a single SNP (see S3 Table).

## GWAS vs. non-GWAS risk prediction performance

One of the motivating factors behind this study was to compare the performance of GWAS-derived or SNP-derived biomarker profiles for disease prediction with other predictive biomarker profiles derived from clinical, metabolomic and/or proteomic (i.e. non-GWAS) data. These data are presented in Table 4 along with types of biomarkers used in each non-GWAS biomarker profile. In compiling this table, we performed an extensive literature survey (of 30 publications) to identify a number of validated or widely used clinical and/or 'omics' biomarker sets for disease risk prediction in 12 different conditions/phenotypes (Alzheimer's, type 2 diabetes, metabolic syndrome, prostate cancer, etc.) along with their reported AUROCs. About one third of the AUROC values quoted in this table used logistic regression while the others used either PLS-DA or random forest classifiers. While the choice of the classification model can affect the reported AUROC, the differences between different types of (good) classifiers are generally small (<5%) [20–23]. The average GWAS AUROC was 0.64 whereas the average non-GWAS AUROC was 0.81.

**Table 4. Comparison of typical metabolic, clinical, proteomic and genetic marker AUROCs for common conditions.**

| Conditions/Phenotypes | Metabolic Marker AUROC [Reference] | Clinical Factor AUROC [Reference] | Proteomic Marker AUROC [Reference] | Genetic Marker AUROC [Reference] |
|---|---|---|---|---|
| Alzheimer's Disease | 0.770 [59] | 0.770 [60] | - - - | 0.640 [61] |
| Type 2 Diabetes | 0.849 [62] | 0.870 [63] | - - - | 0.641 [64] |
| Age Related Macular Degeneration | - - - | 0.780 [65] | - - - | 0.820 [66] |
| Metabolic Syndrome | 0.820 [67] | 0.810 [68] | - - - | 0.640 [69] |
| Colorectal Cancer | 0.895 [70] | - - - | 0.885 [71] | 0.570 [38] |
| Prostate Cancer | - - - | - - - | 0.690 [72] | 0.610 [73] |
| Pulmonary Tuberculosis | - - - | 0.910 [74] | - - - | 0.640 [75] |
| Cardiovascular Disease | - - - | 0.770 [76] | - - - | 0.600 [77] |
| Breast Cancer | - - - | 0.690 [78] | 0.856 [79] | 0.638 [80] |
| Esophageal Squamous-cell Carcinoma | - - - | 0.639 [48] | - - - | 0.632 [48] |
| Leprosy | 0.862 [81] | - - - | - - - | 0.707 [53] |
| Lung Cancer | 0.800 [82] | 0.903 [83] | - - - | 0.551 [47] |
| *Average*: | 0.83 | 0.79 | 0.81 | 0.64 |

## Calculating SNP heritability using the AUROC

While comparing the AUROC values and heritability estimates plotted in the GWAS-ROCS website, an interesting trend was noted. In particular, the heritability seemed to be well correlated with the square of the AUROC (r = 0.87, see Fig 2). This led to a more detailed investigation regarding the potential rational for this observation. Upon further reading we found that the $AUROC = (D + 1)/2$, where D is the Somers' rank correlation [84] between risk profile and disease status (1 = diseased, 0 = not diseased). Note that the squared Somers' D rank correlation is in fact the proportion of explained variance [85], and that the definition of heritability is precisely the proportion of explained variance. Thus, in the context of a SNP-only model

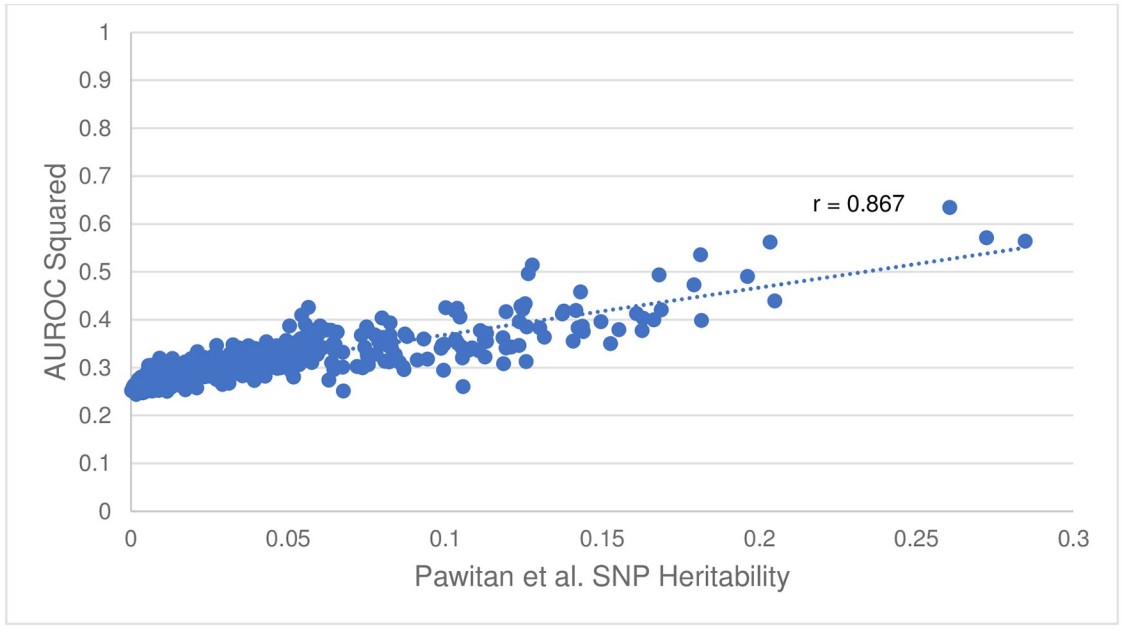

**Fig 2. Plot of the square of the AUROC versus the Pawitan et al. [32] SNP Heritability.** Correlation coefficient, r = 0.867.

trying to predict disease status, the squared Somers' D rank correlation is, in fact, the SNP heritability. Rearranging for D and then squaring in the formula above, we find that $h^2 = D^2 = (2 \cdot AUROC - 1)^2$. This result highlights the utility of AUROC calculations for not only assessing the predictive performance of a multi-SNP panel but for also easily and rapidly calculating heritability of such a SNP panel.

## Comparison with competing methods

As noted earlier, several other methods have been described for estimating AUROC values [8–13] or generating ROC curves [11,12] from summary-level GWAS data. The methods are referred in this paper by the name of the first author. These include the methods by Lu [13], Moonesinghe [9] and Gail [10] which analytically determine AUROC estimates (but not ROC curves), Pepe [12] and Janssens [11] which use population simulation techniques to generate ROC curves from summary-level GWAS data and SummaryAUC [8], a recently released package that can be used to estimate AUROC values based on summary level GWAS data. As noted earlier the methods by Lu, Moonesinghe, Gail and Song do not generate ROC curves nor do they produce population level data. Likewise, the methods by Janssens and Moonesinghe require disease prevalence information to calculate ROC and AUROC data while the method by Janssens is not capable of processing single SNP data. Similarly, the methods by Gail and Lu cannot handle more than 14 SNPs. While SummaryAUC is ideal for calculating AUROCs for a large number of SNPs (100–2000 SNPs) this is not typical of most published GWA studies. As shown in Fig 3, when we compared the actual AUC values generated from the raw WTCCC1 data with the summary-level calculations generated by these programs the performance was disappointing. Indeed, G-WIZ outperforms all of the programs in this test

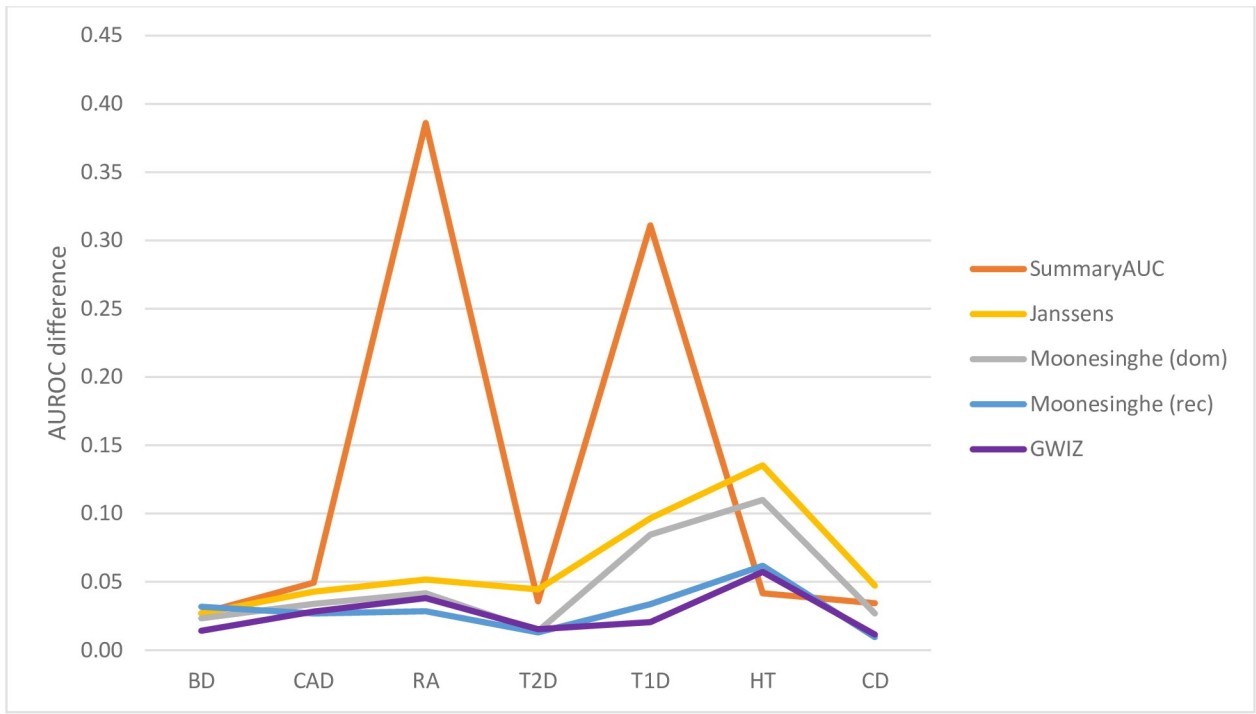

**Fig 3. Comparison of AUROC predictions accuracies made by G-WIZ and SummaryAUC where the difference between the actual AUROC and predicted AUROC for 7 different disorders is shown.** Bipolar disorder (BD), coronary artery disease (CAD), Crohn's disease (CD), hypertension (HT), rheumatoid arthritis (RA), type 1 diabetes (T1D), and type 2 diabetes (T2D).

set. In particular, the average prediction difference was 0.026 for G-WIZ, versus 0.13 for SummaryAUC, 0.033 for Moonesinghe's (dominant inheritance) model, 0.029 for Moonesinghe's (recessive inheritance) model and 0.064 for Janssen's model. Based on these comparisons, we believe that G-WIZ is not only more accurate than other tools or methods so far described, but also requires less input data (no need for disease incidence or frequency). Furthermore, G-WIZ handles single SNP as well as multi-SNP profiles (up to 200 SNPs) and generates a much wider range of useful data including ROC curves, AUROC values, heritability estimates, predictive logistic regression models and simulated patient populations. This diverse output can be used for a much wide number of other research applications that go beyond merely knowing the AUROC.

## The GWAS-ROCS database

Along with the creation of G-WIZ, we also created the GWAS-ROCS Database—a freely available electronic database containing SNP-derived AUROCs calculated via G-WIZ. GWAS-ROCS currently contains all the ROC curves, AUROCs, SNP heritability calculations, simulated populations, references, and SNP meta-data generated in the course of producing this paper. The database currently houses 569 simulated populations (corresponding to 219 different conditions) and SNP data (odds ratios, risk allele frequencies, and p-values) for 2986 unique SNPs. Over the coming months, we intend to extend our web scraping to other databases such as NHGRI-EBI GWAS Catalog. In doing so we expect to be able to expand this database and add dozens more GWAS studies. All the data in the GWAS-ROCS database is downloadable, open-access and intended for applications in genomics, biomarker discovery, and general education.

The GWAS-ROCS database also contains numerous datasets with individual level GWAS simulated populations. A GWAS-ROCS simulated population is a *.csv file with computer-generated "patients" who are marked as either cases or controls. Each individual is assigned data about the presence of risk alleles (1 having the risk allele, or 0 not having the risk allele) at SNPs previously identified as being significant. Users can download these simulated patient populations to develop and test their own genetic risk prediction models or to perform other kinds of synthetic population experiments.

A screenshot montage illustrating the contents and design of the GWAS-ROCS database is shown in Fig 4. As can be seen from this figure the GWAS-ROCS website has a simple webpage layout (Fig 4a). There are four tabs at the top of the page: "Browse Study Simulations", "Downloads", "About" and "Contact Us". Clicking on the "Browse Study Simulations" tab allows users to view a scrollable series of images where they can easily browse through the 569 simulated GWA studies produced in this paper (Fig 4b). Users can sort the GWA studies according to their ID number, the condition/phenotype and the AUROC value. Clicking on a study sends users to a webpage with more information about that specific study (Fig 4c). This includes information such as hyperlinks to the reference GWAS Central study and the original PubMed publication, the number of control and case subjects, the SNP accession IDs, the ORs and RAFs, and the simulated ROC plots, all of which can be found on this page. Additionally, a downloadable *.csv file with the simulated population for that specific GWA study can be found on this page too. The "Downloads" tab gives users a way to quickly and efficiently download the *.csv files with simulated populations and ROC plots, for every single study in GWAS-ROCS (Fig 4d). The "About" tab contains some documentation to help users navigate the site. And finally, the "Contact Us" tab gives users an easy way to contact the GWAS-ROCS team with any questions or concerns they may have.

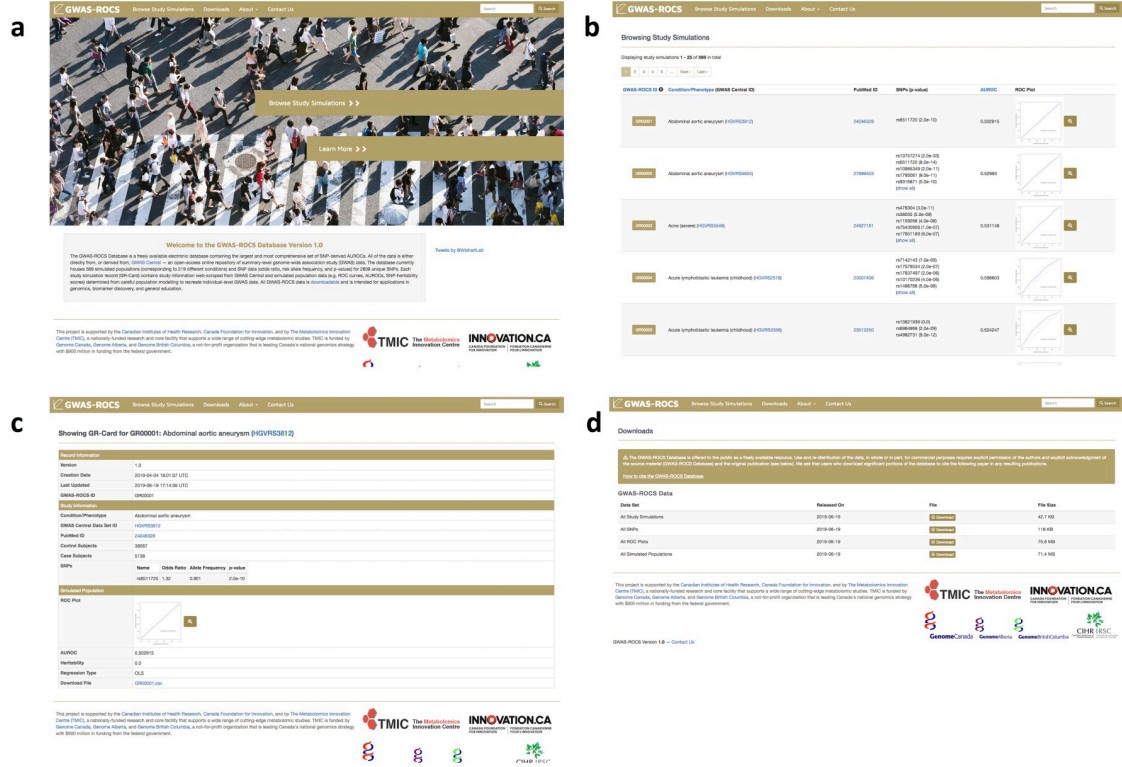

**Fig 4. A screenshot montage of the GWAS-ROCS database at http://gwasrocs.ca. a)** The landing page. **b)** Browse Study Simulations tab. **c)** Study simulation page. **d)** Downloads tab.

## Discussion

GWA studies have contributed significantly to our understanding of the genetic contributions to disease and disease risk. Hundreds of novel genes have been identified and implicated in various traits or conditions and many of these have led to new biological understandings and insights [86]. With continued improvements to GWA study designs (increased sample sizes, better population selection to remove confounders, more narrowly defined phenotypes) and GWAS analysis it is likely that many more important biological or genetic insights will be gained [86]. While the value of GWA studies is indisputable, there are still lingering concerns over the inability of SNPs to explain as much of the heritable variation as originally hoped (the missing heritability problem [87,88]) or as much of the disease risk as expected [88].

As remarked earlier, GWA studies that explore disease risk do not often adopt the convention used by most other multi-marker risk predictors to assess performance. In particular, the use of multi-component SNP models and the evaluation of ROC curves or AUROC values (C-statistics) is quite rare. Of the >3700 GWAS publications we evaluated, only 112 have published ROC curves or AUROC data. Of these, fewer than 30 provided sufficient data to independently validate their reported ROC or AUROC results. This has made it difficult to compare the performance of SNP-derived or GWAS-derived biomarkers in disease risk prediction with other types of disease-risk prediction biomarkers or models (clinical, metabolomic, proteomic, etc.). Furthermore, the difference in reporting methodologies between GWA studies (with an emphasis on p-values and odds ratios for individual SNPs) and non-GWA studies (with an emphasis on ROC curves and AUROCs calculated for multiple markers) has also led to an expectation by many non-GWAS specialists, or those with limited statistical

training, that the predictive performance of GWAS-derived biomarkers should be much better than non-GWAS derived biomarkers.

Because of this "cultural" difference we undertook this study to help standardize biomarker reporting between GWAS derived and non-GWAS derived biomarker profiles. In particular, we used logistic regression modeling and simulated patient data to generate a comprehensive and publicly available database of GWAS ROC curves, AUROCs and SNP-heritability scores for a large number of conditions (219) and a large number of GWAS studies (569). These data were placed into an open-access database called GWAS-ROCS, which is publicly available at http://gwasrocs.ca.

In creating the GWAS-ROCS database we hoped to accomplish several objectives. First, we wanted to compile and consolidate an accurate and comprehensive set of SNP-derived AUR-OCs into a single, open-access site. Second, we wanted to use this consolidated data to systematically analyze interesting trends or features in GWAS AUROC data. One of the trends we wanted to explore in more detail concerned the performance of SNP biomarker panels in disease or phenotype prediction. Our results indicate that the average AUROC for a typical GWAS-derived biomarker profile is low, just 0.55 with a standard deviation of 0.05. This is significantly lower than what we expected given that (the few) published AUROCs typically report a range between 0.62–0.67 (see S2 Table, [88]). The fact that published GWAS AUROCs tend to be high (~0.65) and unpublished GWAS AUROCs tend to be low (~0.55), suggests that one reason for the paucity of published GWAS AUROCs is that many AUROCs for SNP biomarker profiles are either uninterestingly low (<0.55), or not statistically different from those generated by a random predictor.

Another aspect that we wanted to explore in more detail was the performance of GWAS-derived SNP profiles for disease prediction compared to non-GWAS profiles for predicting identical diseases. As noted previously, the fact that GWAS disease risk assessments are not typically presented or measured in the same way as non-GWAS disease risk assessments, has made this kind of comparison difficult [89]. As seen in Table 4, we found that non-genetic factors were generally better at predicting disease than genetic factors. In particular, for those conditions where GWAS-derived AUROCs and non-GWAS (clinical/proteomic/metabolomic) derived AUROCs could be compared, we found that a typical non-GWA study reported AUROCs closer to 0.81, which is significantly more than the average GWAS-derived biomarker profile of 0.64 (see Table 4). On the other hand, it is important to note that the predictive ability or disease risk scores of SNP-derived biomarker profiles can effectively occur at birth (many decades prior to the onset of disease) while the non-SNP-derived biomarkers are generally only useful a few months or at most a few years prior to the onset of the disease. In this regard, the utility of SNP-biomarker profiles for long-term disease prevention or disease prophylaxis, even if modest compared to non-SNP profiles, is still quite significant.

A third objective of this study was to identify those conditions that appear to exhibit the best AUROC performance with multi-component SNP data. These high AUROC conditions would be expected to have a relatively high "explainable" genetic component with regard to disease risk. From the data compiled in GWAS Central we identified 5 conditions/phenotypes that had an AUROC greater than 0.75 and an estimated heritability of >25%. As can be seen in S4 Table, age-related macular degeneration (AUROC = 0.75), celiac disease (AUROC = 0.88), progressive supranuclear palsy (AUROC = 0.81), Craniofacial microsomia (AUROC = 0.76) and black hair colour (AUROC = 0.98) can be largely determined through SNP profiles. It is interesting to note that the very first SNP study ever recorded was one done on macular degeneration [1] and that macular degeneration has among the highest levels of heritability and among the highest AUROC values of all conditions we investigated. In many respects, macular degeneration was the equivalent of hitting the "mother lode" for GWA studies.

A fourth objective of this study was to identify those conditions where SNP information appears to be relatively uninformative with regard to disease risk prediction. Our data indicates that 192 conditions (out of 219) have AUROCs <0.60, and 202 conditions (out of 219) have SNP heritability values of <15%. The majority of these conditions (such as type 1 diabetes, hypertension, amyotrophic lateral sclerosis and Parkinson's disease) are known to have a significant environmental or lifestyle contribution to disease risk. Indeed, non-GWAS derived risk scores (from clinical, proteomic or metabolomic studies) for several of these conditions have predictive AUROCs approaching 0.8 or 0.9. While many GWA studies have been undertaken due to moderately high heritability data estimated from twin studies, it appears that this heritability may be over-estimated due to small sample sizes or through undetected/unaccounted environmental (gut microbiome heritability) or epigenetic effects. We would suggest that prior to conducting large-scale GWA studies based on twin heritability estimates that some assessment regarding disease risk scores with (published) non-GWAS data should be conducted. Objectively assessing the contributions of clinical/phenotypic data, metabolomic data or proteomic data to disease risk prior to conducting a GWA study would certainly identify which kinds of conditions would most likely yield useful GWAS results or useful GWAS-derived biomarkers.

A fifth objective of this study was to demonstrate the utility of using simulated populations to model SNP distributions and to show how these populations, along with logistic regression modeling, could be used to create multi-marker SNP profiles from publicly available, summary-level GWAS data. Critical to this modeling is the assumption that all genotypes and allele proportions were in Hardy-Weinberg equilibrium with no linkage disequilibrium between genes. The method appears to be robust and particularly well-suited to handling most publicly available SNP data. It is also far faster and less resource intensive than having to apply for study access and research ethics approvals (for non-public data) each time a new GWA study is released or a new GWA study is deposited in GWAS Central. However, simulated populations are not "real" and if the exact population and SNP structure is needed to answer a specific question or if absolute precision is needed in determining some SNP-derived model, then there is obviously no substitution for actual clinical data.

A sixth objective of this study was to explore whether certain trends in AUROCs, disease types or heritability estimates could be discerned by analyzing a large AUROC data set. As noted earlier, we found that the SNP-heritability as determined by Pawitan et al. [32], seemed to be well correlated with the square of the AUROC (r = 0.87, see Fig 2). This led to the discovery that $h^2 = (2 \cdot AUROC - 1)^2$. We used this newly derived formula to estimate the SNP-heritability for all 569 studies from GWAS Central. These heritability estimates can be found at http://gwasrocs.ca. Moreover, we compared our heritability estimates against the heritability values for 10 different conditions reported in the literature. Table 5 shows that on average our estimates differed from the true values by just 0.013. The formula we derived suggests that an AUROC of approximately 0.85 is needed to explain 50% of the heritability.

Much is often made of the difference in p-values between GWA studies and the p-values reported for non-GWA studies. As noted before, many GWA studies have SNPs with p-values $< 1 \times 10^{-50}$, while it is rare for non-GWA studies to have clinical, protein or metabolite measures with p-values $<1 \times 10^{-6}$. From our dataset one particular GWA study, "Alzheimer's disease" [97,98], had one SNP with the lowest observed p-value ($1 \times 10^{-295}$) of all SNPs in our GWAS Central dataset. This corresponds to a SNP located close to the well-known Alzheimer's disease-associated gene ApoE4 [99]. Using this single SNP alone it was possible to create an Alzheimer's disease risk predictor with an AUROC of 0.62. The addition of 3 more SNPs with p-values of $2 \times 10^{-10}$, $4 \times 10^{-8}$, $1 \times 10^{-7}$ led to an increase in the Alzheimer's disease risk prediction AUROC to 0.65 [100]. So, despite the extremely low p-values for these Alzheimer's

**Table 5. Comparison of published heritability values versus those predicted using the AUROC for 10 conditions.**

| Condition | Published SNP Heritability | AUROC derived SNP Heritability | Difference |
|---|---|---|---|
| Common psoriasis | 0.280 [90] | 0.270* | 0.010 |
| Crohn's disease | 0.053 [91] | 0.068** | 0.015 |
| Type 1 diabetes | 0.127 [92] | 0.130** | 0.003 |
| Coronary artery disease | 0.106 [93] | 0.073*** | 0.033 |
| Age related macular degeneration | 0.272 [94] | 0.252*** | 0.020 |
| Lupus | 0.209 [95] | 0.213*** | 0.004 |
| Alzheimer's disease | 0.059 [95] | 0.055*** | 0.004 |
| Type 2 diabetes | 0.082 [95] | 0.078*** | 0.004 |
| Inflammatory bowel disease | 0.117 [95] | 0.091*** | 0.026 |
| Rheumatoid arthritis | 0.047 [96] | 0.062*** | 0.015 |
| *Average*: | 0.135 | 0.129 | 0.013 |

* Using AUROC published in the same paper

** Using the true AUROCs calculated from raw WTCCC1 data

*** Using the highest predicted AUROC from GWAS Central

disease SNPs, the influence on the AUROC (and the heritability) was relatively modest. In another interesting example, the GWA study "Coronary Artery Disease" [101,102] had 50 SNPs, and a p-value of a staggering $1 \times 10^{-101}$ for the most significant SNP. However, even with so many SNPs and the inclusion of SNPs with remarkably low p-values, the AUROC of this SNP-panel reached just 0.58. Overall, our results indicate that it is not the p-value, but rather the odds ratio (OR), in conjunction with the risk allele frequency (RAF), that are most important for determining biomarker performance in disease risk prediction.

One criticism of our approach to calculating AUROCs from GWAS data is that is computationally inefficient. In particular, we construct large, simulated patient populations, and then used those simulated patient/SNP populations to estimate the AUROCs. A more efficient approach would have been to use machine learning methods or statistical techniques [9,10] to predict the AUROC values directly. There are two reasons why we chose the population simulation approach. First, we believed it would be more useful for the scientific community to have access to simulated patient populations (with SNP data). This would allow others to perform their own statistical or modeling experiments. Furthermore, simulated SNP data can be used to create synthetic "patients" for electronic medical record (EMR) testing and training. Indeed, because of ethics and privacy restrictions, access to individual level GWAS data is often difficult, making generation of realistic genetic data for patients equally difficult. On the other hand, simulated individual level SNP data (and other 'omics' data) could be of great utility in the development and testing of "next-generation" EMR software and databases with realistic genetic data. As a result, G-WIZ was created as part of a separate EMR project to generate realistic "synthetic" patients with realistic conditions/phenotypes and correspondingly realistic genomic (SNP), metabolomic, proteomic and clinical profiles. In addition to the appeal of creating synthetic patient data, we also realized that by creating simulated populations we would be able to determine and plot ROC curves (with which we could determine the AUROC values). Having a calculated ROC curve would give us another set of data with which to compare and validate our results. Indeed, we used the G-WIZ generated ROC curves to visually validate a number of the early ROC results during the testing phase of the program.

## Conclusion

To summarize, we have created a software tool called G-WIZ to accurately predict GWAS ROC curves and AUROCs from summary level GWAS data. We subsequently compiled data from every sufficiently informative large-scale study in GWAS Central and calculated the corresponding ROC curves and AUROCs using G-WIZ. Using these calculated data, we conducted a number of comparisons to look for interesting results or unexpected trends. In particular, we compared these calculated GWAS AUROCs to typical AUROCs reported in other 'omics' studies and found some striking differences. We also derived a novel formula to calculate SNP-heritability and calculated the proportion of heritability explained by SNPs for all 569 GWAS Central studies that we analyzed. Through this assessment we were able to make some general suggestions regarding the evaluation and selection of medical conditions that should hopefully yield more significant and useful GWAS outcomes. The results of our G-WIZ calculations, along with other meta-data about each GWA study and the predicted heritability have been placed in an open-access database called GWAS-ROCS.

## Supporting information

**S1 Table. SNP heritability calculated using the method described by Pawitan et al. (2009) for every study in GWAS-ROCS.**
(PDF)

**S2 Table. List of every GWA study found in PubMed which reported an AUROC.**
(PDF)

**S3 Table. A breakdown of the number of SNPs reported by the 569 studies we scraped from GWAS Central.**
(PDF)

**S4 Table. List with the highest predicted AUROC for each of the 219 diseases analyzed from GWAS Central.**
(PDF)

## Acknowledgments

The authors thank Mr. John Bacon for his critical comments and suggested edits.

## Author Contributions

**Conceptualization:** Jonas Patron, Arnau Serra-Cayuela, Beomsoo Han, David Scott Wishart.

**Data curation:** Jonas Patron, Carin Li.

**Formal analysis:** Jonas Patron.

**Funding acquisition:** David Scott Wishart.

**Investigation:** Jonas Patron.

**Methodology:** Jonas Patron, Beomsoo Han, David Scott Wishart.

**Project administration:** David Scott Wishart.

**Resources:** David Scott Wishart.

**Software:** Jonas Patron, Arnau Serra-Cayuela, Beomsoo Han, Carin Li.

**Supervision:** David Scott Wishart.

**Validation:** Jonas Patron.

**Visualization:** Jonas Patron, Carin Li.

**Writing – original draft:** Jonas Patron, David Scott Wishart.

**Writing – review & editing:** Jonas Patron, Arnau Serra-Cayuela, Beomsoo Han, Carin Li, David Scott Wishart.

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
