## [Decision Letter · Decision Letter 0]

29 Aug 2019

PONE-D-19-19050

Assessing the performance of genome-wide association studies for predicting disease risk

PLOS ONE

Dear Dr. David,

Thank you for submitting your manuscript to PLOS ONE. After careful consideration, we feel that it has merit but does not fully meet PLOS ONE’s publication criteria as it currently stands. Therefore, we invite you to submit a revised version of the manuscript that addresses the points raised during the review process.

Please address the comments from the reviewer. In addition, please address the review of the program to the general research community.

We would appreciate receiving your revised manuscript by Oct 13 2019 11:59PM. To enhance the reproducibility of your results, we recommend that if applicable you deposit your laboratory protocols in protocols.io, where a protocol can be assigned its own identifier (DOI) such that it can be cited independently in the future. For instructions see: http://journals.plos.org/plosone/s/submission-guidelines#loc-laboratory-protocols

We look forward to receiving your revised manuscript.

Kind regards,

Joseph Devaney

Academic Editor

PLOS ONE

Journal Requirements:

Reviewers' comments:

Reviewer's Responses to Questions

**Comments to the Author**

1. Is the manuscript technically sound, and do the data support the conclusions?

Reviewer #1: Partly

2. Has the statistical analysis been performed appropriately and rigorously? 

Reviewer #1: No

3. Have the authors made all data underlying the findings in their manuscript fully available?

Reviewer #1: Yes

4. Is the manuscript presented in an intelligible fashion and written in standard English?

Reviewer #1: Yes

5. Review Comments to the Author

Reviewer #1: The manuscript presents a statistical method and an R package for approximating the AUC of reported SNPs when only the summary level data of the validation dataset are available. The manuscript is well written, and it is easy to follow.

I have the following major concerns.

On page 7&8, it seems authors first simulate the SNP profile and then assign the healthy/disease statuses. Shouldn’t it simulate the SNP profile given the healthy/disease statues? A detailed explanation is needed for this part.

On page 10, what is the variance inflation factor? How was it used to determine if there is multicollinearity?

On page 14, what are odd/even SNPs?

On page 15, authors claimed that “This was important because the SNP inheritance model in published GWA studies may vary from SNP to SNP, and is not typically disclosed. Thus, for each of the 7 WTCCC disease studies we calculated the true AUROC under these four SNP inheritance schemes.” I do not see the point to have four schemes using logistic and ridge regression. The scheme is automatically handled when the model is being fitted. Why do we need to restrict the inheritance scheme explicitly?

In Table 2, some AUROCs are close to 0.5. Does it mean the report SNPs has little or no contribution to the classification accuracy? Do we have any explanation that low p-value SNPs have little discrimination power?

On page 18, there is a comparison of AUROCs between GWAS-derived vs. non-GWAS-derived risk factors. How were those AUROCs of non-GWAS-derived risk factors calculated? Did they use the same logistic or ridge regression models? If not, the comparison may not be fair. The difference may be caused by the models not the risk factors.

Will the authors release the developed tool as open source? The impact will be limited if the research community cannot verify and apply the computational method.

6. PLOS authors have the option to publish the peer review history of their article (what does this mean?). If published, this will include your full peer review and any attached files.

Reviewer #1: No

---

## [Author Response · Author response to Decision Letter 0]

30 Sep 2019

Sept. 29, 2019

Dear Dr. Devaney,

Please find attached our revised manuscript (PONE-D-19-19050) entitled: “Assessing the performance of genome-wide association studies for predicting disease risk” by Jonas Patron, Arnau Serra-Cayuela, Beomsoo Han, Carin Li and myself. We wish to thank the reviewer for their comments and suggestions. We have tried our best to address all the reviewer’s comments by modifying the paper and have provided detailed responses or described our modifications to the manuscript in the attached pages. We have also produced both a clean version and marked-up version of the manuscript with all changes indicated in red so that the edits or modifications can be more easily seen. 

For Reviewer #1:

1. On page 7&8, it seems authors first simulate the SNP profile and then assign the healthy/disease statuses. Shouldn’t it simulate the SNP profile given the healthy/disease statuses? A detailed explanation is needed for this part.

Response: We appreciate the reviewer’s concerns and apologize for the confusion. Indeed, we simulate the SNP profiles given the healthy/disease status of the cohort. Note from equations (1)-(4) on pg. 7&8 we start by determining or inputting the number of cases and controls. That is, we always start by knowing the healthy/disease status of the individuals in the cohort, we just don’t know the SNP profile of said individuals. Next, by knowing the risk allele frequency in the controls and the odds ratio between the cases and controls we calculate the risk allele frequency in the cases. Once we know the risk allele frequency in both the case and control groups, we appropriately assign the SNP profiles to each group. We have added portions of this text to the manuscript to explain the process in more detail.

2. On page 10, what is the variance inflation factor? How was it used to determine if there is multicollinearity?

Response: Given a regression model with multiple variables, the variance inflation factor is the quotient of the variance from a model which regresses one of the predictor variables against all the others. Multicollinearity was determined to exist when at least two variables showed inflated coefficients, as quantified by the calculation of the variance inflation factor being equal to infinity. We also tried other variance inflation factor cutoff values, however because the differences in the AUC estimates were so small (<0.009) and because standard logistic regression is more easily interpretable than its ridge regression counterpart, we found it appropriate to restrict the use of ridge regression only to models with extreme (i.e. divergent) variance inflation factor estimates. We have added portions of this text to the manuscript to explain this concept in more detail.

3. On page 14, what are odd/even SNPs?

Response: The characterization of SNPs as odd or even is simply referring to their position in a list—those found at odd numbered positions in the list are “odd SNPs” and those that are found in even numbered positions are called “even” SNPs. It is a simple scheme to split the SNPs in an approximately 50-50 fashion. Consider Table R1 (see below), which is a table with SNPs. The 1st, 3rd, and 5th SNPs would be the odd SNPs and the 2nd and 4th SNPs would be the even SNPs.

Table R1

Accession Control size Case size Risk allele freq OR Odd/Even

rs7903146 2598 2309 0.3 1.36 odd

rs5219 2598 2309 0.36 1.25 even

rs10811661 2598 2309 0.85 1.21 odd

rs1801282 2598 2309 0.87 1.21 even

rs2641348 2598 2309 0.11 1.15 odd

4. On page 15, authors claimed that “This was important because the SNP inheritance model in published GWA studies may vary from SNP to SNP, and is not typically disclosed. Thus, for each of the 7 WTCCC disease studies we calculated the true AUROC under these four SNP inheritance schemes.” I do not see the point to have four schemes using logistic and ridge regression. The scheme is automatically handled when the model is being fitted. Why do we need to restrict the inheritance scheme explicitly?

Response: This is an excellent point. What were doing was conducting two different and independent modeling steps. In the first step, we were trying to calculate the true AUROC of a given WTCCC disease study. To do this we had to convert the raw GWAS data into dummy variables to which we could apply logistic regression. Take Table R2 (below) for example. In this example, there is one individual and we have information about four of his/her SNPs. To run a regression analysis the information in columns “1st Allele” and “2nd Allele” must be converted to a single number (either a 0 or 1) known as the dummy variable. Then, in the second step we would run our logistic/ridge regression to estimate the AUROC (in this step the choice between logistic or ridge regression would be handled automatically). Note that in this simple example a change in the choice of the SNP inheritance scheme produces three distinct results: (1,1,0,1), (0,1,0,0), and (1,1,0,0). As a result, when we run our standard regression techniques we could expect to produce 3 different AUROCs, all of which can be interpreted as the “true” AUROC.

Table R2

SNP Risk Allele 1st Allele 2nd Allele Dominant Inheritance Recessive Inheritance Dominant (odd) – Recessive (even) 

#1 A T A 1 0 1

#2 A A A 1 1 1

#3 T A A 0 0 0

#4 C C G 1 0 0

As such, to rigorously test GWIZ’s AUROC predictions, we had to ensure that we could accurately predict the AUROC which resulted from any one of these inheritance schemes. In our test set we could restrict the inheritance scheme explicitly, however in practice, given a set of odds ratios we cannot necessarily know which inheritance scheme produced them.

5. In Table 2, some AUROCs are close to 0.5. Does it mean the report SNPs has little or no contribution to the classification accuracy? Do we have any explanation that low p-value SNPs have little discrimination power?

Response: With regard to the first question, yes, when the AUROCs are close to 0.5 it does indicate that the reported SNPs has/have little or no contribution to the classification accuracy. With regard to the second question, the size of the p-value and its influence on the AUROC is primarily affected by the associated SNP’s odds ratio and allele frequency. In other words, low p-values can arise from statistically significant effects of small magnitude. In GWA studies, the ‘effect’ we are looking for is that of differences in allele frequencies between case and control groups. And the magnitude of the effect in question is measured by the odds ratios. A SNP with a p-value of 10-80 but with an odds ratio close to 1 (which is typical of most GWAS SNPs) will have a very small effect on the AUROC. A low odds ratio often arises from a very low SNP/allele frequency. A SNP with an odds ratio of 3 but a p-value of just 10-10 will have a large effect on the AUROC. 

6. On page 18, there is a comparison of AUROCs between GWAS-derived vs. non-GWAS-derived risk factors. How were those AUROCs of non-GWAS-derived risk factors calculated? Did they use the same logistic or ridge regression models? If not, the comparison may not be fair. The difference may be caused by the models not the risk factors.

Response: The AUROCs of the non-GWAS derived risk factors were calculated using a mixture of different approaches. About 1/3 of the values quoted in this table used logistic regression while the others used either PLS-DA or random forest methods. It is true that the choice of the classification model can affect the performance and the reported AUROC but generally most biostatisticians will test multiple classification models (logistic regression, PLS-DA, random forest, SVM regression, etc.) and will quote the best performing model. Our own experience in using many different classifiers for many other related projects is that the AUROC differences between classifiers are generally quite small (<5%) although rare exceptions do occur. Overall, we believe the AUROC differences accurately reflect the risk factors and that they are not a (significant) function of the choice of classifier. We have added this information to the manuscript.

7. Will the authors release the developed tool as open source? The impact will be limited if the research community cannot verify and apply the computational method.

Response: The code is freely available for download at https://bitbucket.org/wishartlab/gwiz-rscript/src/master/, and is compatible with a wide variety of UNIX platforms, Mac OS and Windows operating systems. The above link to bitbucket is cited in the original manuscript on page 11, however to reduce confusion in the future we have also mentioned this link in the abstract.

In addition to these edits we have also made some small changes to the manuscript by adding a few missing references and elaborating on GWIZ’s comparison to other programs. This led to a small update to Figure 3. Once again, we would like to thank the reviewer for their comments and suggestions. We hope that the explanations we have provided are satisfactory and certainly the changes they suggested have significantly improved the overall clarity and quality of the manuscript. We hope these edits are acceptable and are looking forward to seeing the manuscript published in PLOS ONE soon.

Sincerely,

David Wishart

---

## [Decision Letter · Decision Letter 1]

4 Nov 2019

Assessing the performance of genome-wide association studies for predicting disease risk

PONE-D-19-19050R1

Dear Dr. David,

We are pleased to inform you that your manuscript has been judged scientifically suitable for publication and will be formally accepted for publication once it complies with all outstanding technical requirements.

With kind regards,

Joseph Devaney

Academic Editor

PLOS ONE

Additional Editor Comments (optional):

Reviewers' comments:

Reviewer's Responses to Questions

**Comments to the Author**

1. If the authors have adequately addressed your comments raised in a previous round of review and you feel that this manuscript is now acceptable for publication, you may indicate that here to bypass the “Comments to the Author” section, enter your conflict of interest statement in the “Confidential to Editor” section, and submit your "Accept" recommendation.

Reviewer #1: All comments have been addressed

2. Is the manuscript technically sound, and do the data support the conclusions?

Reviewer #1: Yes

3. Has the statistical analysis been performed appropriately and rigorously? 

Reviewer #1: Yes

4. Have the authors made all data underlying the findings in their manuscript fully available?

Reviewer #1: Yes

5. Is the manuscript presented in an intelligible fashion and written in standard English?

Reviewer #1: Yes

6. Review Comments to the Author

Reviewer #1: (No Response)

7. PLOS authors have the option to publish the peer review history of their article (what does this mean?). If published, this will include your full peer review and any attached files.

Reviewer #1: No

---

## [Editor Report · Acceptance letter]

21 Nov 2019

PONE-D-19-19050R1 

Assessing the performance of genome-wide association studies for predicting disease risk 

Dear Dr. David:

I am pleased to inform you that your manuscript has been deemed suitable for publication in PLOS ONE. Congratulations! Your manuscript is now with our production department. 

With kind regards,

on behalf of

Dr. Joseph Devaney 

Academic Editor

PLOS ONE